# Ideology Takes Multiple Looks: A High-Quality Dataset for Multifaceted Ideology Detection

**Songtao Liu**[1,*] **Ziling Luo**[2,*] **Minghua Xu**[2,†] **Lixiao Wei**[1],
**Ziyao Wei**[2]**, Han Yu**[1]**, Wei Xiang**[1]**, Bang Wang**[1,†]

[1]School of Electronic Information and Communications,
Huazhong University of Science and Technology, Wuhan, China
[2]School of Journalism and Information Communication,
Huazhong University of Science and Technology, Wuhan, China
{liusongtao, luoziling, xuminghua, wangbang}@hust.edu.cn

## Abstract

Ideology detection (ID) is important for gaining insights about peoples' opinions and stances on our world and society, which can find many applications in politics, economics and social sciences. It is not uncommon that a piece of text can contain descriptions of various issues. It is also widely accepted that a person can take different ideological stances in different facets. However, existing datasets for the ID task only label a text as ideologically left- or right-leaning as a whole, regardless whether the text containing one or more different issues. Moreover, most prior work annotates texts from data resources with known ideological bias through distant supervision approaches, which may result in many false labels. With some theoretical help from social sciences, this work first designs an ideological schema containing five domains and twelve facets for a new *multifaceted ideology detection* (MID) task to provide a more complete and delicate description of ideology. We construct a MITweet dataset for the MID task, which contains 12,594 English Twitter posts, each annotated with a Relevance and an Ideology label for all twelve facets. We also design and test a few of strong baselines for the MID task under *in-topic* and *cross-topic* settings, which can serve as benchmarks for further research. [1]

## 1 Introduction

Ideologies are the collection of beliefs or philosophies attributed to a particular social group or class (Eagleton, 2007). They shape how we see the world and interact with each other. In this information age, social media has rapidly developed, allowing people to express their thoughts and opinions online, which offer strong cues about their ideologies (Preoţiuc-Pietro et al., 2017; Kulkarni

---

*   Equal contribution: S. Liu and Z. Luo.
†   Corresponding author: M. Xu and B. Wang

[1]Dataset and codes are available on `https://github.com/LST1836/MITweet`

| Authors | Source | Size | Annotated Manually | Multi-faceted |
|---|---|---|:---:|:---:|
| Iyyer et al. (2014) | Books | 3,412 | ✓ | ✗ |
| Preoţiuc-Pietro et al. (2017) | Twitter | 4,833,133 | ✗ | ✗ |
| Kulkarni et al. (2018) | News articles | 120,000 | ✗ | ✗ |
| Kiesel et al. (2019) | News articles | 1,273 | ✓ | ✗ |
| Kiesel et al. (2019) | News articles | 754,000 | ✗ | ✗ |
| Baly et al. (2020) | News articles | 34,737 | ✓ | ✗ |
| Liu et al. (2022) | News articles | 2,331,552 | ✗ | ✗ |
| García-Díaz et al. (2022) | Twitter | 37,560 | ✗ | ✗ |
| MITWEET | Twitter | 12,594 | ✓ | ✓ |

Table 1: Comparison of ideology detection datasets.

et al., 2018; Schwarz, 2019). Ideology detection is a critical task for quantitative political and social science (Wilkerson and Casas, 2017; Liu et al., 2022), which can help policy makers to analyze public opinions for making wise decisions (Xiao et al., 2020). Tracking ideology on social media is also important for monitoring online communities and detecting signs of ideological radicalization (Grover and Mark, 2019; Aldera et al., 2021).

It is widely accepted that individuals or groups can take different ideological stances in different facets (Boyd and Jackson, 1967; Feldman and Huddy, 2014). Motivated by this, many efforts have been devoted to characterizing ideology from various facets (Ferguson, 1952; Rokeach, 1973; Grunow et al., 2018). From a computational linguistics viewpoint, some texts may contain descriptions of different issues and reflect the author's ideology from various aspects. As shown in Figure 1(b), the authors convey their multiple ideologies by expressing opinions on topics they care about. Therefore, it is necessary to detect the ideology of texts from multiple facets, so as to provide a more comprehensive and nuanced picture for further sociological research.

In order to facilitate progress on ideology detection, a number of annotated datasets have been

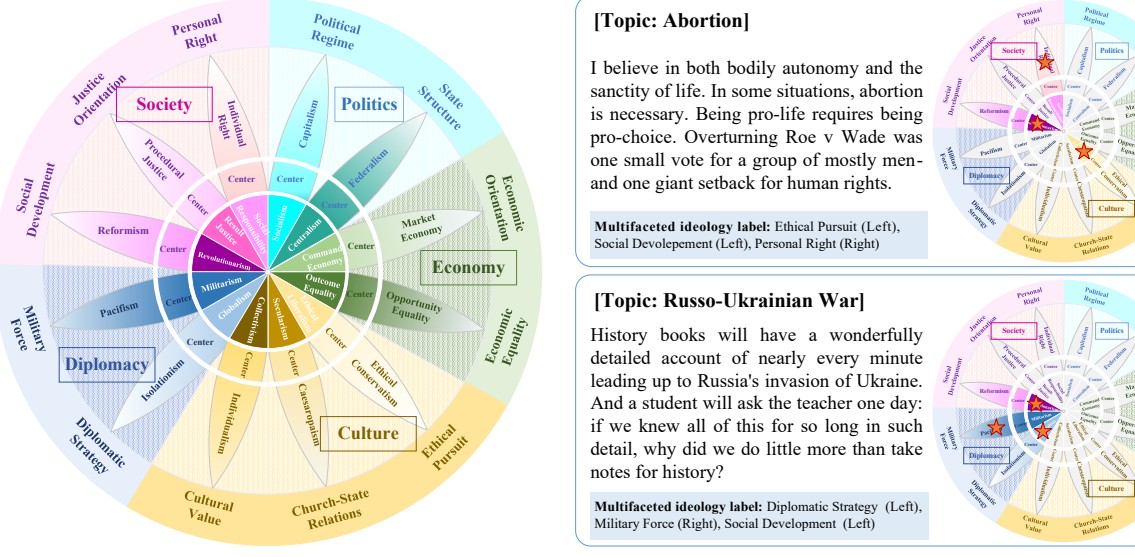

(a) Our proposed multifaceted ideology schema.

(b) Examples from our MITweet dataset.

Figure 1: (a) An illustration of our proposed multifaceted ideology schema. Five fans represent the five domains and twelve petals represent the facets under domains. The inside and outside of each petal denote the left-leaning and right-leaning ideologies of that facet respectively. (b) Examples from our MITweet dataset. Pentagrams are used to mark the multifaceted ideology label. Colored petals represent the facets that the text involves.

released, as shown in Table 1. However, two drawbacks of existing datasets limit the research of this task: (1) Only one facet. Due to the absence of theoretical guidance from the multifaceted ideology, existing datasets only label a text as ideologically left- or right-leaning as a whole, regardless whether the text containing one or more different issues. For example, the upper text in Figure 1(b) can be labeled with left-leaning due to its radical attitude. However, this simple label conceals the domains discussed in the text (i.e., *culture* and *society*), and ignores that the author supports the protection of individual rights, which is conservative or right-leaning in sociology. (2) Noisy labels. Most of previous work crawl data from resources with known ideological bias and project the bias label to all texts gathered from it, which is a form of distant supervision. As discussed in Baly et al. (2020), while the distant supervision assumption generally holds, there are still many instances that defy it, which introduces noise into datasets.

In an effort to address these issues, in this work, we first design a multifaceted ideology schema to provide a more complete and delicate description of ideology, by considering traditional ideology division and emerging social topics. As shown in Figure 1(a), the schema covers five orthogonal domains, including politics, economy, culture, diplomacy and society, under which, there are a total

of twelve facets. We also define the ideological attributes for the left- and right-leaning of each facet to achieve a more detailed illustration of ideology. Based on the new schema, we construct a new high-quality dataset, MITweet, for a new *multifaceted ideology detection* (MID) task. The dataset contains 12,594 English Twitter posts, each manually annotated with a *Relevance* (and an *Ideology*) label for all twelve facets. As the examples in Figure 1(b), MITweet makes up the absence of comprehensive annotation by annotating ideological labels for multiple facets involved in texts. In addition, MITweet covers 14 highly controversial topics in recent years (e.g., abortion, covid-19 and Russo-Ukrainian war), which can facilitate research in cross-topic MID.

Using MITweet as the benchmark, we develop strong baselines for the new MID task based on several widely-used pre-trained language models. We conduct thorough experiments and analysis under both in-topic and cross-topic settings. Experiment results show that: (1) BERTweet (Nguyen et al., 2020) performs the best on overall metrics for both subtasks of MID. (2) The performance of BERTweet can be further improved by using *indicators* as part of input, which is detected with the weighted log-odds-ratio technique with informed Dirichlet priors (Monroe et al., 2008). (3) Cross-topic setting is quite challenging and the achieved

performances are far from promising.

## 2 Multifaceted Ideology

To accurately measure the ideology of an individual or group, sociologists have designed a series of scales to categorize political attitudes based on one or more facets. Ferguson (1952) developed ten stance scales that encompass attitudes towards birth control, the death penalty, censorship, communism, evolution, law, patriotism, theism, criminals, and war. By carefully analyzing the results of the popular test, he was able to condense the ten scales into three distinct ideological facets: religious, humane, and ethnic. Soon after, Hans (1957) classified political ideology into "Radicalism" and "Conservatism". To demonstrate the inadequacy of the above research, Rokeach (1973) proposed a dual-axis model of ideology based on the principles of freedom and equality. This model is similar to the Nolan Chart (Nolan, 1976), a political diagram that uses two facets to divide people's ideological positions into four different quadrants. The horizontal axis represents "economic freedom", while the vertical axis represents "individual freedom". Ideologies located in different quadrants are categorized as modern liberalism, conservatism, libertarianism, and nationalism. These prior work proposed different facets for dividing ideologies and laid the theoretical foundation for multifaceted ideology, but a core problem is that they usually study specific domains, such as politics or economics, and different facets may overlap or have ambiguous boundaries.

With the development of society, the study of multifaceted ideology and political spectrum has brought attention to new issues, including anti-war movements, women's rights, environmental protection, animal welfare and other related concerns. Mueller (2017) argued that color-blind racism is a form of covert and highly institutionalized discrimination, which differs from the more overt forms of racism seen during slavery and legal segregation. Therefore, colorblindness serves as the prevailing ideological support for a historically unique form of structural white supremacy known as color-blind racism. Grunow et al. (2018) argued that defining gender ideology as a one-dimensional construct that ranges from traditional to egalitarian is problematic in contemporary society. They proposed an alternative framework that considers the multidimensionality of gender ideologies and identified

five ideology profiles: egalitarian, essentialist egalitarian, intensive parenting, moderately traditional, and traditional. During the COVID-19 pandemic, the far-right ideology has taken on new characteristics, including a disregard for scientific expertise, distrust of news media, and allegiance to Trump-Christian nationalism (Perry et al., 2020). These new social issues call for a flexible ideological evaluation system, which can fully explain contemporary issues in various social fields and describe the existing variation in ideology from multiple facets.

By considering previous work and emerging social issues, we put forth a new *multifaceted ideology schema* (see Figure 1(a), detailed in Appendix A). Firstly, our multifaceted ideology schema divides ideology into five orthogonal domains to ensure that each domain is independent and has clear boundaries. Secondly, the multifaceted ideology schema now incorporates new contemporary issues. For instance, the facet "Cultural Value" is linked to the anti-abortion movements, the facet "Justice Orientation" is associated with the Black Lives Matter movement, and the facet "Military Force" is connected to peace-war issues. Finally, our multifaceted ideology schema covers twelve facets, each is defined with ideological attributes for the left- and right-leaning, providing a more complete and delicate description of ideology.

## 3 The MITweet Dataset

Based on the proposed multifaceted ideology schema as described in Section 2, we annotate tweets manually to build MITweet. In this section, we describe how the data is collected, the design of annotation process, dataset statistics and the methods to ensure the quality of annotation.

### 3.1 Data Collection

We collect tweets using Twitter streaming API based on different topics. To maximize the proportion of tweets containing the author's ideology and improve annotation efficiency, we select 14 topics that meet the following two criteria: (1) highly controversial or extensively discussed in recent years; (2) related to politics, economy, culture, diplomacy or society, i.e. the five domains in the ideology schema. Then we use a set of query keywords as seeds to collect topic-related tweets. For each topic, we set the query period during which the event involved many intensive discussions on social media,

| Topic | Query Keywords | Query Period | #Raw Tweets | #Cleaned Tweets | #Annotated Tweets |
|---|---|---|---|---|---|
| Inflation Reduction Act (IRA) | #IRA, #inflation | Aug - Sep, 2022 | 1114 | 811 | 427 |
| Capitol Hill Riot (CHR) | #CapitolHillRiot | Jan - Feb, 2021 | 2330 | 1440 | 346 |
| CHIPS and Science Act (CSA) | #CHIPSAct | Aug 2022 | 2159 | 1469 | 346 |
| Abortion (Ab) | #RoeVWade | Jun - Aug, 2022 | 3222 | 2376 | 1893 |
| Russo-Ukrainian War (RUW) | #UkraineWar, Kyiv | Feb - Oct, 2022 | 7390 | 5672 | 2100 |
| Mass Shootings (MS) | #GunsShot, #crime | Feb 2018 - Jun 2022 | 2325 | 1494 | 624 |
| George Floyd (GF) | #GeorgeFloyd | May - Aug, 2020 | 896 | 582 | 423 |
| Black Lives Matter (BLM) | #BlackLivesMatter | Aug 2014 - Jun 2020 | 3877 | 1026 | 707 |
| Political Parties (PP) | Democrat, Republican | Mar 2020 - Oct 2022 | 1692 | 1253 | 502 |
| Mexico–US Border (MUB) | #BorderCrisis | Mar - May, 2021 | 3337 | 2331 | 912 |
| Democracy (Dm) | democracy | Sep - Nov, 2022 | 829 | 750 | 392 |
| Covid-19 (C19) | covid, vaccine | Feb 2020 - Jul 2022 | 3718 | 3070 | 369 |
| Energy Crisis (EC) | #EnergyCrisis, #oil | Mar - Apr, 2022 | 2205 | 1523 | 661 |
| Women's Right (WR) | women right | Dec 2021 - May 2022 | 6405 | 5170 | 2892 |
| **Total** | \ | \ | **41499** | **28967** | **12594** |

Table 2: 14 topics in MITweet. The left part presents examples of query keywords and the query periods when collecting data. The right part presents the numbers of raw tweets, cleaned tweets after preprocessing and final annotated tweets for each topic.

as shown in Table 2. In total, we gather around 41.5k raw tweets for all topics.

To ensure the quality of MITweet, we preform several preprocessing steps for raw tweets: (1) We remove duplicates and retweets. (2) We remove @USER and URLs in tweets, as they often do not convey semantic information literally. (3) We remove tweets with less than 15, or more than 130 words. We set this length limit because tweets that are too short are usually semantically incomplete, while tweets that are too long may contain many irrelevant content, which can introduce noise into the dataset. (4) We focus on tweets that have received a lot of attention on Twitter. Tweets that express a clear or controversial point of view or represent the perspectives of many people often have higher popularity and research value.

For each tweet, we calculate a spread score based on the number of likes, comments and retweets, a user score based on the number of tweets, followers and followees of the author. Then a final heat score is obtained by weighting the spread score and user score. We remove tweets with heat scores below a certain threshold. After preprocessing, we have 29.0k cleaned tweets in total and randomly sample 17.0k of them for annotation.

## 3.2 Data Annotation

We invite 56 annotators to participate in our annotation, including graduate students from schools of public management, social science and communications. Before starting the annotation, we conduct sufficient training for annotators and several rounds of annotation trials, which will be discussed in detail in Section 3.4.

For each tweet, each annotator needs to annotate a *Relevance* (and an *Ideology*) label for each of twelve facets in our ideology schema described in Section 2. This process was done in two steps. Firstly, annotators are asked to label "Related" or "Unrelated" for each facet. For example, if an annotator believes that a tweet expresses the author's ideology regarding "Social Development", then "Related" should be labeled on this facet, and vice verse. Secondly, for each facet labeled with "Related" in the previous step, annotators are further required to label one ideological category: "Left", "Center" or "Right". Note that the "Center" class means that tweets are biased towards a centrist ideology in one facet, not lacking ideological bias (e.g., purely descriptive news reports), which is labeled with "Unrelated".

Each tweet is annotated by three random annotators, and the final results are obtained through major voting. If major voting is not possible, that is, three annotators label a facet with completely different ideological categories, then the tweet will be discarded due to its possible high ambiguity. When only two annotators label "Related" on a facet, they must reach an agreement on ideological

| Domain | Facet | Relevance | | Ideology | | |
|---|---|---|---|---|---|---|
| | | #Related | Related Ratio (%) | #Left | #Center | #Right |
| Politics | Political Regime (PoR) | 112 | 0.9 | 39 | 14 | 59 |
| | State Structure (SS) | 291 | 2.3 | 67 | 88 | 136 |
| Economy | Economic Orientation (EO) | 759 | 6.0 | 294 | 297 | 168 |
| | Economic Equality (EE) | 672 | 5.3 | 520 | 119 | 33 |
| Culture | Ethical Pursuit (EP) | 2935 | 23.3 | 1976 | 465 | 494 |
| | Church-State Relations (CSR) | 68 | 0.5 | 33 | 17 | 18 |
| | Cultural Value (CV) | 154 | 1.2 | 95 | 11 | 48 |
| Diplomacy | Diplomatic Strategy (DS) | 1572 | 12.5 | 711 | 421 | 440 |
| | Military Force (MF) | 1837 | 14.6 | 132 | 575 | 1130 |
| Society | Social Development (SD) | 1737 | 13.8 | 1236 | 287 | 214 |
| | Justice Orientation (JO) | 3452 | 27.4 | 3058 | 281 | 113 |
| | Personal Right (PeR) | 3516 | 27.9 | 171 | 241 | 3104 |

Table 3: Relevance and ideology distribution. Related ratio refers to the proportion of tweets related to a facet in the entire MITweet.

labels; Otherwise, the tweet will also be discarded. Finally, 74.1 percent of labeled tweets pass the inter-annotator agreement check and compose the MITweet dataset.

### 3.3 Dataset Statistics and Analysis

**Overall Statistics** MITweet contains a total of 12,594 tweets, 11,649 of which are related to at least one ideological facet in the schema. Each tweet is labeled with a *Relevance* label, and an *Ideology* label if the *relevance* label is "Related", along each facet. Meanwhile, MITweet involves 14 topics from a variety of fields, and their statistics are provided in Table 2 and Appendix C.

**Relevance Distribution** As shown in Table 3, PeR, JO and EP are the most frequently related facets; MF, SD and DS are also comparatively common, while CSR and PoR are the rarest facets, with a related ratio of less than 1%. In general, most facets have a relatively low related ratio, resulting in an imbalance data distribution and posing challenges for facet relevance detection task. Nevertheless, we do not take downsampling since this is a reflection of the real-world data distribution.

**Ideology Distribution** The right part of Table 3 presents the ideology statistics. It is obvious that different facets have very different data distributions. For example, most tweets are left-leaning towards SD and JO, while it is the opposite for MF and PeR. This once again indicates that it is

not appropriate to label texts just as either left- or right-leaning and that characterizing ideology from multiple facets can provide a more complete and nuanced view for further analysis.

### 3.4 Quality Control

As described in Section 1, most previous related work annotate datasets using distant supervision approaches, which can lead to many noisy labels in datasets. In contrast, our MITweet is annotated manually and is therefore inherently more reliable and diverse.

In order to further ensure the data quality, we carried out a strict annotation workflow. First of all, we explain the annotation schema in detail for annotators and provide some examples. Then several rounds of annotation trials are performed. After each trial, we collect questions from annotators and discuss frequent inconsistencies, based on which we retrain annotators and then a new round of trial will start. The above process is iterated for several times until the pass rate of agreement check reaches 0.70 and the average inter-annotator agreement reaches 0.75 (Krippendorff's alpha). During formal annotation, we annotate 14 topics one by one to avoid interference between topics and reduce the difficulty in dealing with continuously changing topics. Before starting the annotation of each topic, we introduce the background of the topic and provide some representative instances for annotators.

| Facet | PoR | SS | EO | EE | EP | CSR | CV | DS | MF | SD | JO | PeR | **Avg.** |
|---|---|---|---|---|---|---|---|---|---|---|---|---|---|
| **K's alpha** | 92.8 | 88.1 | 86.1 | 77.5 | 80.3 | 89.3 | 93.0 | 90.2 | 71.6 | 84.0 | 68.6 | 75.7 | **83.1** |

Table 4: The Krippendorff's alpha of each ideology facet.

We compute the Krippendorff's alpha along each facet as the measure of agreement between annotators, as shown in Table 4. We can observe that Krippendorff's alpha of most facets are above 80, with an average of 83.1, which is a satisfactory score, indicating high consistency among annotators and that the dataset is reliable for further research.

### 3.5 Credibility Analysis

With the well-designed schema and strict annotation workflow, the data distribution of MITweet is generally consistent with the sociological characteristics of each topic involved. Take the topic of BLM as an example.

BLM (Black Lives Matter) is a social and civil rights movement. It advocates for the rights and equality of Black communities in various aspects of society, including law enforcement, criminal justice, education, employment and healthcare. From the perspective of multifaceted ideology, BLM is mainly related to Social Development, Justice Orientation, Personal Right (Society domain) and Ethical Pursuit (Culture domain). In addition, the nature of BLM, which is the pursuit of black rights, racial equality and social reform, suggests that it stands for Revolutionism (Left) in Social Development, Result Justice (Left) in Justice Orientation, Individual Right (Right) in Personal Right and Ethical Liberalism (Left) in Ethical Pursuit.

The topic distribution of MITweet is shown in Appendix C, we can see that the label distribution of BLM topic is consistent with the analysis mentioned above. The facets with the largest number of related tweets are Justice Orientation, Social Development, Personal Right and Ethical Pursuit, accounting for 88.4%, 36.5%, 23.9% and 5.4% of all BLM tweets, respectively. The ideology distribution also aligns with the sociological characteristics of BLM. Most tweets related to Ethical Pursuit (81.6%), Social Development (80.6%) and Justice Orientation (89.0%) are left-leaning in their respective facets. And 87.6% of tweets related to Personal Right are right-leaning in this facet.

For other topics, we can observe similar consistency with sociological analysis. This indicates that MITweet reflects people's ideological leanings towards various facets in the real world and therefore has high credibility.

## 4 Experiments

### 4.1 Dataset Split

We split the MITweet dataset in two different ways for different scenarios. **(1) Random split.** We randomly divide the training, development, and testing sets following a 70/15/15 split. This is for in-topic MID since all three sets share the same topics. **(2) Topic split.** Firstly, several topics are selected as the test topics. Then, tweets of the remaining topics are randomly divided into training set and development set according to the ratio of 4:1. This can serve as the setting of cross-topic MID, where the model is first trained and validated on source topics, and then tested on target topics.

### 4.2 Tasks and Models

We split the multifaceted ideology detection procedure into two sub-tasks in a pipeline manner: **(1) Relevance Recognition** aims to recognize the facets that a text is related to. We model this task as a multi-label classification problem. **(2) Ideology Detection** predicts which ideology a text holds regarding the related facets. This task can be modeled as a multi-class classification problem.

We use Transformer (Vaswani et al., 2017) encoder based pre-trained language models (PLMs) as backbone models. Three PLMs are compared in our experiments: BERT (Devlin et al., 2019), RoBERTa (Liu et al., 2019), and BERTweet (Nguyen et al., 2020) pretrained on 850M English Tweets following the training procedure of RoBERTa. The base version of these models from HUGGINGFACE are used. We fine-tune the PLMs to predict the *relevance* or *ideology* by appending a linear classification layer on top of the hidden representation of the [CLS] token. For the *relevance recognition task*, tweet text is the input. For the *ideology detection task*, we concatenate the tweet with the facet name (e.g., "Political Regime") and separate them using the [SEP] token, and then feed the sequence into PLMs.

| Facet | Indicators |
|-------|-----------|
| PoR | Socialism, Communist, capitalism |
| SS | Democrats, Republicans, partisan |
| EO | #inflation, #IRA, market |
| EE | insurance, relief, tax |
| CSR | Christian, religious, church |
| JO | women, #BlackLivesMatter, rights |

Table 5: Examples of indicators.

Inspired by Kawintiranon and Singh (2021), which used the weighted log-odds-ratio technique with informed Dirichlet priors (Monroe et al., 2008) to compute stance words, we propose using the similar method to obtain the *indicators* for each facet. Specially, we extract tweets from training set to form a corpus $\mathcal{C}_i$ and a corpus $\mathcal{C}_j$, where $\mathcal{C}_i$ contains tweets related to a facet, while $\mathcal{C}_j$ contains the rest, i.e., unrelated tweets. Then, by using weighted log-odds-ratio, we compute the usage difference of each word among two corpora and find the top-$k$ significant words in $\mathcal{C}_i$ as *indicators* of the facet. Therefore, *indicators* are the words in training set that best represent a facet. Examples of indicators are provided in Table 5. Obviously, compared with the abstractness of facet names, indicators have more concrete meanings and different indicators describe different perspectives of a facet. We set $k = 18$ and concatenate tweet with *indicators* as an alternative input for the *ideology detection task*.

### 4.3 Evaluation Metrics

We adopt the Accuracy (Acc) and F1 metrics for each facet. In order to evaluate the overall performance of models, we use both *micro* and *macro* methods to integrate metrics from each facet. Macro-Acc/F1 treats each facet equally and is the average of Acc/F1 scores across all facets. Micro-Acc/F1 is calculated globally by putting together predictions from all facets. Note that due to the highly unbalanced data distribution in the *relevance recognition task*, we only use F1-related metrics in this task for more credible evaluations.

### 4.4 Results and Analysis

#### 4.4.1 In-topic Setting

Under in-topic setting, we train, validate and test models on the same 14 topics in MITweet. F1 scores of each facet are shown in Table 6. We also provide Acc results of the *ideology detection task* for each facet in Appendix D.

For the *relevance recognition task*, first, we can observe that BERTweet performs the best on most facets and achieves competitive performance on EP and SD, indicating the effectiveness of this model, which is pretrained on tweets. Second, facets with low results, such as CV, SS, CSR and PoR, are also the facets with the most unbalanced data distributions as provided in Section 3.3. This demonstrates that dealing with the low proportion of positive instances in this task is a huge challenge and needs more research efforts. One exception is the SD facet, which has a similar rated ratio as DS and MF, but much worse performance. By analyzing the prediction results, we find that many SD-related tweets use metaphors and hide the intent behind the literal sense(e.g., the lower example in Figure 1(b)), which is difficult for models to understand.

For the *ideology detection task*, we can observe that: (1) The achieved performances are low across all facets, with no facet exceeding 60%, which is far from practically usable, indicating the inherent challenge of this task. (2) Different from the *relevance recognition task*, in this task, each model has its own advantageous facets. For example, BERTweet leads other models by a large margin in CSR, especially after using indicators, while BERT and RoBERTa exhibit the best performances in five facets respectively. (3) The performances of eight facets are improved when using *indicators* as part of input. This is likely due to the fact that indicators come from the words that appear in the dataset and may contain more topic-related in-contextual meanings compared with bare facet names. That is, indicators can establish a deeper interaction with texts and help models mine the ideological bias contained in texts more accurately.

Overall results for relevance recognition and ideology detection are presented in Table 7 and Table 8 respectively. BERTweet achieves the best overall performance on both sub-tasks, except for Macro-F1 in the *ideology detection tasks*, which is also competitive with the best result. This is not unexpected, because BERTweet is pre-trained on tweets and is therefore more suitable for modeling the tweet texts in this task, whereas BERT and RoBERTa have domain adaptation problems to some extent. Interestingly, after using *indicators*, we can see a significant improvement on BERTweet, but not for BERT and RoBERTa. We think this is likely because indicators contain some words with Twitter style (e.g., *#inflation*, *#Black-*

| | PoR | SS | EO | EE | EP | CSR | CV | DS | MF | SD | JO | PeR |
|---|---|---|---|---|---|---|---|---|---|---|---|---|
| | | | | *Relevance Recognition* | | | | | | | | |
| BERT | **50.05**$_{7.21}$ | 30.91$_{4.61}$ | 67.16$_{2.43}$ | 61.59$_{1.45}$ | **82.64**$_{0.98}$ | 34.07$_{6.55}$ | 8.46$_{4.08}$ | 61.06$_{3.52}$ | 85.72$_{0.67}$ | 42.10$_{1.48}$ | 74.59$_{1.68}$ | 69.38$_{0.85}$ |
| RoBERTa | 44.55$_{4.86}$ | 31.76$_{5.47}$ | 69.21$_{1.85}$ | 61.04$_{1.56}$ | 81.55$_{1.26}$ | **39.89**$_{4.55}$ | 18.39$_{5.06}$ | 62.08$_{3.25}$ | 85.10$_{0.91}$ | **44.54**$_{2.77}$ | 75.13$_{1.47}$ | 69.76$_{0.87}$ |
| BERTweet | 46.92$_{2.59}$ | **32.71**$_{1.38}$ | **71.05**$_{0.69}$ | **63.29**$_{0.95}$ | 82.26$_{0.46}$ | 35.04$_{2.61}$ | **19.52**$_{2.43}$ | **62.73**$_{1.55}$ | **85.99**$_{1.84}$ | 44.07$_{2.70}$ | **75.55**$_{1.24}$ | **70.71**$_{0.74}$ |
| | | | | *Ideology Detection* | | | | | | | | |
| *Facet Name* | | | | | | | | | | | | |
| BERT | 26.93$_{6.30}$ | 24.16$_{3.70}$ | 53.28$_{3.54}$ | **48.51**$_{1.05}$ | 49.90$_{1.28}$ | 33.05$_{14.41}$ | 44.97$_{8.70}$ | **57.59**$_{0.53}$ | 46.37$_{2.32}$ | 42.32$_{1.43}$ | 37.18$_{1.12}$ | 36.31$_{3.09}$ |
| RoBERTa | 28.38$_{11.46}$ | 29.61$_{8.12}$ | 51.84$_{2.27}$ | 46.96$_{8.63}$ | **53.66**$_{2.69}$ | 31.65$_{8.89}$ | 47.56$_{16.09}$ | 56.09$_{1.81}$ | 47.41$_{3.30}$ | 42.63$_{2.16}$ | 37.44$_{1.71}$ | **40.20**$_{5.67}$ |
| BERTweet | 27.40$_{10.54}$ | 28.97$_{4.61}$ | 54.17$_{3.21}$ | 47.98$_{8.65}$ | 51.36$_{0.99}$ | 36.42$_{9.06}$ | 46.15$_{13.80}$ | 57.23$_{1.73}$ | 44.03$_{1.55}$ | 42.88$_{4.14}$ | 36.51$_{1.30}$ | 34.06$_{3.04}$ |
| *Indicators* | | | | | | | | | | | | |
| BERT | **31.79**$_{9.27}$ | 29.93$_{4.20}$ | **54.42**$_{4.97}$ | 46.58$_{5.27}$ | 51.44$_{2.64}$ | 19.96$_{7.22}$ | 44.74$_{9.07}$ | 54.62$_{3.45}$ | 47.11$_{2.05}$ | 44.08$_{2.97}$ | **38.83**$_{1.21}$ | 40.04$_{5.22}$ |
| RoBERTa | 23.45$_{3.46}$ | **33.50**$_{5.30}$ | 52.18$_{4.25}$ | 43.74$_{8.28}$ | 52.41$_{3.31}$ | 28.17$_{9.37}$ | **51.77**$_{11.01}$ | 56.03$_{1.91}$ | 46.88$_{3.84}$ | **45.00**$_{2.37}$ | 35.31$_{1.55}$ | 39.34$_{5.33}$ |
| BERTweet | 24.40$_{2.69}$ | 27.59$_{5.07}$ | 52.26$_{4.34}$ | 41.25$_{8.06}$ | 52.43$_{2.79}$ | **49.93**$_{6.07}$ | 43.37$_{11.81}$ | 57.00$_{2.59}$ | **48.39**$_{6.02}$ | 43.92$_{2.63}$ | 36.55$_{2.53}$ | 35.04$_{2.40}$ |

Table 6: F1 scores (%) of the two sub-tasks for each facet under in-topic setting. *Facet Name* and *Indicators* denote the two types of input described in Section 4.2. **Bold** and underlined values are the best and second-best results in the sub-task, respectively. We report the average of 5 runs along with their standard deviation in the subscript.

| | Macro-F1 | Micro-F1 |
|---|---|---|
| BERT | 55.64$_{0.58}$ | 69.36$_{0.39}$ |
| RoBERTa | 56.92$_{0.92}$ | 69.87$_{0.58}$ |
| BERTweet | **57.48**$_{0.53}$ | **70.32**$_{0.43}$ |

Table 7: Overall results (%) of relevance recognition.

| | Macro-F1 | Micro-F1 | Macro-Acc | Micro-Acc |
|---|---|---|---|---|
| *Facet Name* | | | | |
| BERT | 41.72$_{1.83}$ | 66.39$_{0.67}$ | 62.50$_{1.10}$ | 74.53$_{1.11}$ |
| RoBERTa | **42.78**$_{3.34}$ | 67.80$_{1.11}$ | 63.75$_{1.26}$ | 75.59$_{1.00}$ |
| BERTweet | 42.27$_{1.68}$ | 68.42$_{1.38}$ | 64.51$_{1.74}$ | 75.10$_{1.61}$ |
| *Indicators* | | | | |
| BERT | 41.96$_{1.10}$ | 66.48$_{1.28}$ | 62.35$_{1.35}$ | 74.26$_{1.89}$ |
| RoBERTa | 41.78$_{3.38}$ | 67.63$_{0.63}$ | 64.71$_{1.43}$ | 75.98$_{1.04}$ |
| BERTweet | 42.68$_{2.52}$ | **69.28**$_{0.65}$ | **65.88**$_{1.45}$ | **76.38**$_{0.42}$ |

Table 8: Overall results (%) of ideology detection.

| Target Topics | Relevance Recognition | Ideology Detection | |
|---|---|---|---|
| | Micro-F1 | Micro-Acc | Micro-F1 |
| CHR&GF | 59.60$_{0.30}$ | 70.20$_{0.85}$ | 52.41$_{0.81}$ |
| BLM&Dm | 54.69$_{0.29}$ | 80.64$_{0.42}$ | 58.89$_{0.47}$ |

Table 9: Zero-shot results (%). We report averages along with their standard deviations over 5 random test sets from target topics.

*LivesMatter*), which is common during BERTweet pre-training, but may confuse BERT and RoBERTa. Moreover, the Macro- metrics are always significantly lower than the Micro- metrics, indicating that the models are good at making overall predictions across all facets, but not ideal at distinguishing between different facets.

### 4.4.2 Cross-topic Setting

New topics come up every day, but acquiring large amounts of annotated texts for new topics is time-consuming and labor-intensive. It is hence necessary to evaluate the models' generalization ability to transfer knowledge from annotated topics. We next conduct two sets of experimentations with BERTweet under cross-topic setting: **(1) Zero-shot.** A model is first trained and validated on source topics, and then tested on target topics. **(2) Few-shot.** In addition to texts of source topics, a model can be further trained on a few samples from each target topic, with the parameters of PLM frozen or not. We ensure that the test sets of few-shot and zero-shot are the same, for a fair comparison.

Most of the topics are domain-specific, and it is likely that there is little or no relevant text on some facets. For example, topic BLM has no relevant texts in EO and CSR, and only one or two relevant texts in PoR, SS, DS and MF, as shown in Appendix C. Therefore, we only use micro-Acc and micro-F1 metrics to focus on facets that are more relevant to a topic.

Zero-shot results are provided in Table 9. We can observe that the Micro-Acc of ideology detection is relatively high and even reaches 80.64% when BLM&Dm are used as target topics, while the Micro-F1 of both sub-tasks are rather low. This indicates that the model tends to predict majority categories and does not have a good understanding of the relevance and ideology features of texts. The main reason is that under cross-topic setting,

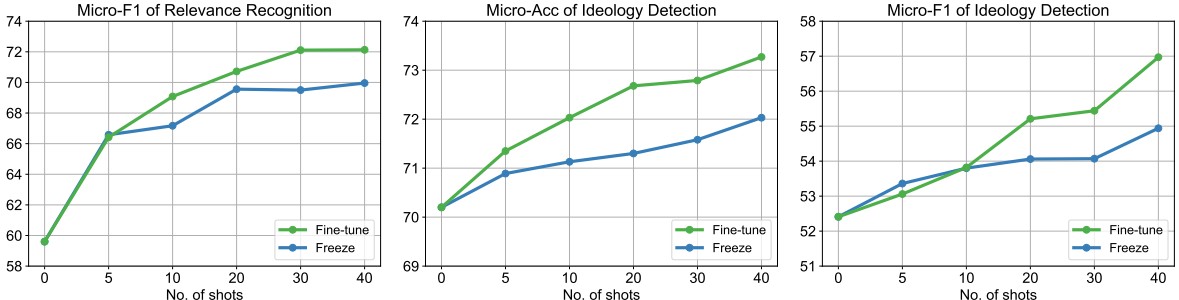

Figure 2: Few-shot results (%). Target topics are CHR&GF. "Fine-tune" and "Freeze" respectively denote that we fine-tune and freeze the parameters of PLMs during model training. We report averages over 5 random training and test sets from target topics.

models must be able to discover the association between source and target topics to gain better performance. However, without any knowledge about the target topics, models can only rely on connections on text level, so it is hard to correctly identify the relevance and ideology.

In Figure 2, we present the results for few-shot experiments. We can see a clear upward trend in performance as the training data increases, and almost all results under the *fine-tune* setting are better than those under the *freeze* setting. For the *relevance recognition task*, the model gains considerable improvements with only 5 training examples. When the number of training examples increases to 40, there is a large gap of about 10 compared with the results in zero-shot. However, for the *ideology detection task*, the performance improves slightly even in 40-shot, suggesting that, it is still hard for the model to transfer ideological features from source topics with few training examples and maybe there is a need to incorporate knowledge about target topics into the model in future work.

## 5 Conclusion

In this paper, we have presented a new multifaceted ideology schema covering five domains and twelve facets to provide a more complete and delicate evaluation system for ideology. Based on the schema, we have constructed a high-quality dataset, MITweet, as a benchmark for a new MID task. Experiments show that the MID task is quite challenging, especially under cross-topic setting, which requires the knowledge transfer capability of models. We believe that this work has the potential to positively impact both the research and the applications of ideology detection.

## Limitations

- MITweet only covers English tweets, which limits the linguistic features covered by MITweet and the scope of applications built on it. In some facets (e.g., CSR and PoR), the number of relevant texts is relatively small and may not train the model well. Therefore, we will annotate more texts with the schema by combining emerging new topics and extend MITweet to other languages in future work.

- Only several base PLM models are chosen in our experiments, and more ideology detection methods are worth of further exploring.

- Under cross-topic setting, we just train models with texts from source topics. As the experimental analysis shows, it is worth considering injecting prior knowledge of target topics to improve the generalization ability of models. We leave this work for the future.

## Ethics Statement

The proposed multifaceted ideology schema is in accordance with the general ethics in social science research to understand people, society and politics. The MITweet dataset is only a peek into the huge amount of Twitter data, and the released dataset has removed all identity related information.

## Acknowledgement

This work is supported in part by Major Project of National Social Science Foundation of China: "AI and Precise International Communication" (Grant No. 22&ZD317) and National Natural Science Foundation of China (Grant No. 62172167). We thank all the annotators for their hard work.

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

## A Multifaceted Ideology Schema

The multifaceted ideology schema contains *five domains* that reflect different aspects of society. Under the five domains, there are *twelve facets* with ideological attributes of left- and right-leaning, as shown in Table 10. Detailed definitions are as follows.

### A.1 Domain 1: Politics

**Political Regime (PoR)**    The formal and informal structure and nature of political power in a country.

- *Socialism* (Left): the ownership or control of the property should be public-owned.

- *Capitalism* (Right): the ownership of the means of production should private-owned.

**State Structure (SS)**    The organizational form of the state, i.e., the distribution of power among agencies.

- *Centralism* (Left): the power should be concentrated in the central authority.

- *Federalism* (Right): the power should be distributed between a central authority and the constituents.

### A.2 Domain 2: Economy

**Economic Orientation (EO)**    Any of the ways in which humankind has arranged for its material provisioning.

- *Command Economy* (Left): the government should take responsibility for making most of the important economic decisions.

- *Market Economy* (Right): economic decisions should be guided by the interactions of individuals, organization, companies.

**Economic Equality (EE)**    The orientation that ought to be adopted in achieving equality through economic policies.

- *Outcome Equality* (Left): all groups should receive the same treatment or distribution.

- *Opportunity Equality* (Right): all groups should have equal access to resources.

### A.3 Domain 3: Culture

**Ethical Pursuit (EP)**    The guiding principles and standards that govern the collective existence and conduct of individuals.

- *Ethical Liberalism* (Left): the mainstream culture should support sexual liberation, same-sex marriage, abortion, and other related issues.

- *Ethical Conservatism* (Right): the mainstream culture should restrict individuals' behavior based on moral norms and religious doctrines.

**Church-State Relations (CSR)**    The relations between religious ideology and the political consciousness of a nation.

- *Secularism* (Left): the religious power and state power should be separated.

- *Caesaropapism* (Right): the religious power and state power should be unified.

| Domain | Facet | Left | Right |
|--------|-------|------|-------|
| Politics | Political Regime (PoR)
State Structure (SS) | Socialism
Centralism | Capitalism
Federalism |
| Economy | Economic Orientation (EO)
Economic Equality (EE) | Command Economy
Outcome Equality | Market Economy
Opportunity Equality |
| Culture | Ethical Pursuit (EP)
Church-State Relations (CSR)
Cultural Value (CV) | Ethical Liberalism
Secularism
Collectivism | Ethical Conservatism
Caesaropapism
Individualism |
| Diplomacy | Diplomatic Strategy (DS)
Military Force (MF) | Globalism
Militarism | Isolationism
Pacifism |
| Society | Social Development (SD)
Justice Orientation (JO)
Personal Right (PeR) | Revolutionism
Result Justice
Social Responsibility | Reformism
Procedural Justice
Individual Right |

Table 10: The proposed multifaceted ideology schema.

**Cultural Value (CV)**   The cognitive framework shared by the members of society.

- *Collectivism* (Left): an individual should be seen as subordinate to a social collective.

- *Individualism* (Right): all values should be human-centred and the individual should be of supreme importance.

### A.4   Domain 4: Diplomacy

**Diplomatic Strategy (DS)**   Fundamental principles and guidelines for a nation's diplomatic endeavors.

- *Globalism* (Left): foreign policy should be planned with an international perspective.

- *Isolationism* (Right): political and economic entanglements with other countries should be avoided.

**Military Force (MF)**   The disposition of a nation or political faction towards the utilization of military force.

- *Militarism* (Left): it is necessary to use strong armed forces to gain political or economic advantages.

- *Pacifism* (Right): all types of violence between countries are incorrect.

### A.5   Domain 5: Society

**Social Development (SD)**   The collective stance of societal members towards the flux of eras.

- *Revolutionism* (Left): it is necessary to take direct and noticeable action to achieve social goals.

- *Reformism* (Right): the social changes should take place in a gradual way.

**Justice Orientation (JO)**   The orientation that ought to be adopted in achieving justice through social policies.

- *Result Justice* (Left): people should be fairly distributed and treated in various social activities.

- *Procedural Justice* (Right): the authorities should make fair decisions.

**Personal Right (PeR)**   The standards by which individual rights are measured in the formulation of social policies.

- *Social Responsibility* (Left): there should be a greater emphasis on fulfilling individual responsibilities.

- *Individual Right* (Right): there should be a greater emphasis on protecting individual rights.

## B   Calculation Process of Tweet Heat Score

For each tweet, the spread score is calculated by weighting the number of likes $L$, comments $C$ and retweets $R$. We also consider the influence

of the tweet author and calculate the user score by weighting the number of tweets $T$, followers $Fr$ and followees $Fe$ of the author. Then the final heat score is obtained by weighting the spread score and user score.

$$\text{spread} = L * \alpha_1 + C * \alpha_2 + R * \alpha_3,$$
$$\text{user} = T * \beta_1 + Fr * \beta_2 + Fe * \beta_3,$$
$$\text{heat} = \text{spread} * \mu_1 + \text{user} * \mu_2.$$

During data collection, we set $\alpha_1 = 0.3, \alpha_2 = 0.6, \alpha_3 = 0.1, \beta_1 = 0.3, \beta_2 = 0.6, \beta_3 = 0.1$ and $\mu_1 = 0.6, \mu_2 = 0.4$. We remove tweets with heat scores below a threshold which is adjusted manually according to the topic heat on Twitter.

## C  Topic Distribution of MITweet

See Table 11.

## D  Additional Experimental Results

See Table 12.

| | Topic | IRA | CHR | CSA | Ab | RUW | MS | GF | BLM | PP | MUB | Dm | C19 | EC | WR |
|---|---|---|---|---|---|---|---|---|---|---|---|---|---|---|---|
| | #Tweets | 427 | 346 | 346 | 1893 | 2100 | 624 | 423 | 707 | 502 | 912 | 392 | 369 | 661 | 2892 |
| | #Reated Tweets | 407 | 330 | 319 | 1890 | 1995 | 242 | 411 | 699 | 447 | 837 | 379 | 330 | 531 | 2832 |
| PoR | #Left | 0 | 2 | 2 | 1 | 6 | 0 | 0 | 0 | 3 | 2 | 9 | 4 | 6 | 4 |
| | #Center | 1 | 0 | 2 | 0 | 3 | 0 | 0 | 1 | 3 | 2 | 2 | 0 | 0 | 0 |
| | #Right | 1 | 0 | 1 | 3 | 8 | 1 | 0 | 0 | 22 | 6 | 7 | 6 | 2 | 2 |
| SS | #Left | 4 | 0 | 1 | 3 | 2 | 0 | 0 | 0 | 49 | 3 | 4 | 0 | 0 | 1 |
| | #Center | 4 | 1 | 0 | 7 | 1 | 1 | 0 | 1 | 56 | 7 | 9 | 0 | 0 | 1 |
| | #Right | 6 | 2 | 0 | 21 | 6 | 0 | 2 | 1 | 63 | 4 | 20 | 5 | 0 | 6 |
| EO | #Left | 112 | 0 | 30 | 0 | 5 | 0 | 0 | 0 | 27 | 3 | 2 | 7 | 108 | 0 |
| | #Center | 115 | 1 | 26 | 0 | 12 | 1 | 0 | 0 | 22 | 1 | 1 | 8 | 107 | 3 |
| | #Right | 63 | 0 | 11 | 0 | 4 | 0 | 0 | 0 | 7 | 1 | 1 | 1 | 80 | 0 |
| EE | #Left | 78 | 3 | 8 | 30 | 10 | 5 | 1 | 7 | 96 | 34 | 8 | 134 | 50 | 56 |
| | #Center | 41 | 0 | 4 | 2 | 1 | 0 | 0 | 2 | 29 | 8 | 0 | 17 | 10 | 5 |
| | #Right | 11 | 0 | 0 | 2 | 1 | 0 | 0 | 1 | 8 | 6 | 1 | 2 | 1 | 0 |
| EP | #Left | 1 | 6 | 0 | 1268 | 2 | 13 | 0 | 31 | 7 | 2 | 5 | 3 | 1 | 637 |
| | #Center | 0 | 2 | 0 | 297 | 3 | 50 | 2 | 4 | 2 | 4 | 1 | 0 | 0 | 100 |
| | #Right | 0 | 34 | 1 | 252 | 2 | 107 | 10 | 3 | 5 | 16 | 3 | 7 | 0 | 54 |
| CSR | #Left | 0 | 2 | 0 | 16 | 0 | 1 | 0 | 0 | 0 | 0 | 3 | 0 | 0 | 11 |
| | #Center | 0 | 0 | 0 | 7 | 2 | 0 | 0 | 0 | 1 | 0 | 0 | 1 | 0 | 6 |
| | #Right | 0 | 3 | 0 | 7 | 1 | 1 | 0 | 0 | 3 | 0 | 0 | 1 | 0 | 2 |
| CV | #Left | 3 | 0 | 1 | 4 | 4 | 2 | 1 | 4 | 36 | 3 | 5 | 29 | 1 | 2 |
| | #Center | 0 | 0 | 0 | 0 | 3 | 1 | 0 | 0 | 3 | 0 | 2 | 1 | 0 | 1 |
| | #Right | 5 | 1 | 0 | 15 | 1 | 1 | 0 | 6 | 10 | 2 | 0 | 1 | 1 | 5 |
| DS | #Left | 7 | 3 | 109 | 0 | 383 | 4 | 1 | 1 | 2 | 101 | 14 | 14 | 68 | 4 |
| | #Center | 2 | 1 | 62 | 0 | 193 | 0 | 1 | 1 | 5 | 112 | 5 | 12 | 25 | 2 |
| | #Right | 1 | 1 | 53 | 0 | 55 | 0 | 0 | 0 | 3 | 271 | 1 | 12 | 43 | 0 |
| MF | #Left | 1 | 0 | 3 | 0 | 111 | 1 | 0 | 0 | 2 | 2 | 1 | 0 | 11 | 0 |
| | #Center | 2 | 0 | 5 | 0 | 546 | 1 | 0 | 0 | 0 | 3 | 0 | 1 | 17 | 0 |
| | #Right | 0 | 3 | 6 | 2 | 1069 | 4 | 1 | 1 | 2 | 5 | 3 | 3 | 27 | 4 |
| SD | #Left | 41 | 104 | 24 | 125 | 13 | 18 | 122 | 208 | 51 | 114 | 120 | 65 | 63 | 168 |
| | #Center | 27 | 67 | 5 | 42 | 4 | 6 | 12 | 32 | 8 | 8 | 20 | 10 | 12 | 34 |
| | #Right | 17 | 101 | 0 | 24 | 0 | 1 | 2 | 18 | 9 | 10 | 6 | 0 | 4 | 22 |
| JO | #Left | 8 | 34 | 6 | 147 | 17 | 26 | 352 | 556 | 24 | 149 | 58 | 35 | 7 | 1639 |
| | #Center | 1 | 5 | 1 | 13 | 2 | 7 | 17 | 50 | 6 | 23 | 9 | 2 | 0 | 145 |
| | #Right | 0 | 1 | 1 | 1 | 1 | 2 | 5 | 19 | 4 | 34 | 5 | 0 | 0 | 40 |
| PeR | #Left | 3 | 15 | 4 | 26 | 1 | 9 | 6 | 19 | 10 | 13 | 14 | 22 | 0 | 29 |
| | #Center | 0 | 2 | 0 | 61 | 4 | 3 | 2 | 2 | 7 | 5 | 16 | 6 | 0 | 133 |
| | #Right | 5 | 23 | 5 | 532 | 21 | 29 | 69 | 148 | 24 | 101 | 175 | 39 | 14 | 1919 |

Table 11: Topic distribution of MITweet. "Related Tweets" means tweets related to at least one facet in our schema.

| | PoR | SS | EO | EE | EP | CSR | CV | DS | MF | SD | JO | PeR |
|---|---|---|---|---|---|---|---|---|---|---|---|---|
| *Facet Name* | | | | | | | | | | | | |
| BERT | $42.50_{7.29}$ | $35.29_{3.22}$ | $58.91_{2.11}$ | $76.91_{1.54}$ | $70.21_{1.32}$ | $38.18_{10.60}$ | $59.17_{4.86}$ | $\mathbf{60.34}_{1.65}$ | $65.41_{2.35}$ | $67.57_{4.42}$ | $\underline{88.46}_{1.34}$ | $87.07_{1.51}$ |
| RoBERTa | $38.75_{13.35}$ | $38.24_{7.21}$ | $58.18_{3.15}$ | $74.85_{2.49}$ | $70.72_{2.40}$ | $38.18_{6.80}$ | $\mathbf{70.00}_{8.08}$ | $59.23_{1.96}$ | $\mathbf{70.22}_{1.29}$ | $69.14_{5.13}$ | $88.39_{0.87}$ | $\mathbf{89.05}_{0.63}$ |
| BERTweet | $42.50_{8.29}$ | $\underline{40.59}_{4.33}$ | $\mathbf{61.45}_{2.48}$ | $77.11_{3.48}$ | $70.49_{3.54}$ | $\underline{47.27}_{3.64}$ | $63.33_{8.90}$ | $58.80_{2.22}$ | $67.26_{1.76}$ | $70.21_{4.77}$ | $87.73_{0.94}$ | $87.34_{1.19}$ |
| *Indicators* | | | | | | | | | | | | |
| BERT | $\mathbf{48.75}_{7.29}$ | $38.82_{3.43}$ | $59.82_{3.61}$ | $73.81_{3.95}$ | $69.05_{2.79}$ | $25.45_{8.91}$ | $\mathbf{64.17}_{6.77}$ | $57.52_{3.80}$ | $67.34_{1.77}$ | $68.57_{4.83}$ | $86.31_{4.03}$ | $\underline{88.62}_{0.67}$ |
| RoBERTa | $\underline{43.75}_{11.18}$ | $\mathbf{42.36}_{3.99}$ | $59.82_{2.78}$ | $\underline{77.32}_{0.92}$ | $\mathbf{72.53}_{1.35}$ | $40.00_{4.45}$ | $\mathbf{64.17}_{7.73}$ | $59.06_{2.83}$ | $\underline{69.33}_{1.00}$ | $\underline{72.86}_{2.81}$ | $87.50_{2.32}$ | $87.84_{0.80}$ |
| BERTweet | $\underline{43.75}_{6.85}$ | $35.30_{9.84}$ | $\underline{60.91}_{0.58}$ | $\mathbf{77.53}_{1.51}$ | $\underline{71.97}_{1.34}$ | $\mathbf{58.19}_{4.45}$ | $63.33_{3.12}$ | $\underline{59.32}_{1.41}$ | $68.82_{0.72}$ | $\mathbf{74.71}_{2.48}$ | $\mathbf{88.85}_{0.63}$ | $87.92_{0.94}$ |

Table 12: Acc scores (%) of the ideology detection sub-task for each facet under in-topic setting.