# OpenReview forum: "Ideology Takes Multiple Looks: A High-Quality Dataset for Multifaceted Ideology Detection"
_EMNLP/2023/Conference — EMNLP 2023 Main_

### Official Review · Reviewer_upds · 2023-08-05

**Soundness:** 4

**Excitement:**

4: Strong: This paper deepens the understanding of some phenomenon or lowers the barriers to an existing research direction.

**Paper Topic And Main Contributions:**

This paper construct a MITweet dataset for the multifaceted ideology detection(MID) task, which contains 12,594 English Twitter posts each annotated with a Relevance and an Ideology label for all twelve facets. It also design and test a few of baselines for the MID task under in-topic and cross-topic settings, which can serve as benchmarks for further research. This work is sound and interesing, which is beneficial for the reserarcher on fine-grained ideology detection.

**Reasons To Accept:**

(1) The main strength lies in its novel approach to ideology detection through a multifaceted lens. The authors effectively present the concept of exploring ideologies from multiple perspectives, which is crucial for gaining a more comprehensive understanding of this complex and multifaceted phenomenon.
(2) The idea of creating a high-quality dataset to capture these nuances is commendable and fills a gap in the existing research landscape.

**Reasons To Reject:**

no

**Reproducibility:**

5: Could easily reproduce the results.

**Reviewer Confidence:**

5: Positive that my evaluation is correct. I read the paper very carefully and I am very familiar with related work.

---

> ### Author Rebuttal · Authors · 2023-08-28
>
> **We greatly appreciate your thoughtful review and recognition of our work!**

---

### Official Review · Reviewer_bU4Y · 2023-08-07

**Soundness:** 3

**Excitement:**

4: Strong: This paper deepens the understanding of some phenomenon or lowers the barriers to an existing research direction.

**Paper Topic And Main Contributions:**

The authors propose a new idealogy identification schema for annotating ideologies in social media text moving beyond the traditional left and right labels. They create a new dataset called MITweet based on tweets on multiple topics annotated with this new idealogy schema. The authors also ran experiments using different text encoders to present a reasonable baseline for the dataset

**Reasons To Accept:**

New annotation schema which moves beyond binary labels and provides more fine-grained information
Baseline results
New benchmark


**Reasons To Reject:**

Lack of limitations of the new schema
The multi-faceted schema significantly overlaps with topical identification e.g. Politics, Econoics, etc this can make the downstream value of this schema less relevant.


**Reproducibility:**

4: Could mostly reproduce the results, but there may be some variation because of sample variance or minor variations in their interpretation of the protocol or method.

**Reviewer Confidence:**

4: Quite sure. I tried to check the important points carefully. It's unlikely, though conceivable, that I missed something that should affect my ratings.

---

> ### Author Rebuttal · Authors · 2023-08-28
>
> We appreciate your thoughtful review. We address the comments as follows.
>
> >**Q1: Lack of limitations of the new schema. The multi-faceted schema significantly overlaps with topical identification e.g. Politics, Economics, etc this can make the downstream value of this schema less relevant.**
>
> **A1:** This is an important concern. However, we believe that although the 5 domains and 12 facets in our schema may be regarded as topics in topical identification to some extent, our schema still has unique scientific research value and application prospects in the field of social science and computational linguistics.
>
> First, the 5 domains and 12 facets in the new schema are not randomly chosen, but specially designed for multifaceted ideology detection with theoretical research in social sciences, as described in Section 2. We refer to a series of scales that categorize political attitudes based on one or more facets (e.g., the dual-axis model of ideology). We also consider the characteristics of contemporary social issues (e.g., the gender ideology in women’s rights) to describe the existing variation in ideology. In the new schema, the 5 domains reflect different aspects of society and the 12 facets are all issues of great concern in contemporary society in their respective domains. (We will add the explicit definitions of 12 facets in next paper revision.) As a result, our new schema avoids the ambiguity and incompleteness in previous works, and is more suitable for contemporary issues.
>
> Second, for each of the 12 facets, we define the ideological attributes for left- and right-leaning to provide a more precise understanding for measuring multifaceted ideology. In contrast, such definition of ideological attributes does not exist in topical identification.
>
> Third, with this new multifaceted ideology schema, we are able to measure ideology in a more complete and delicate way. As described in Section 1, some texts may contain descriptions for different issues and reflect the author’s ideology from various aspects. Therefore, it is necessary to detect the ideology of texts from multiple facets, so as to provide a more comprehensive and nuanced picture for further sociological research. To our best knowledge, we are the first to propose a multifaceted ideology schema and detect ideology from multiple facets in the field of computational linguistics.

---

### Official Review · Reviewer_4GHc · 2023-08-09

**Soundness:** 3

**Excitement:**

4: Strong: This paper deepens the understanding of some phenomenon or lowers the barriers to an existing research direction.

**Paper Topic And Main Contributions:**

The author proposed  a  multifaceted ideology schema  that covers five domains and twelve facets to provide a complete evaluation system for ideology. Based on the schema, they constructed a high-quality dataset known as  MITweet, as a benchmark for a new multifaceted ideology detection (MID) task.

**Questions For The Authors:**

How do you avoid/ reduce the bias when  measuring the ideology?
Are all the 56 annotators  in the article from graduate school?

**Reasons To Accept:**

The baselines used  for the MID task under in-topic and cross-topic settings.

**Reasons To Reject:**

Ideologies are often complex and can have multiple dimensions. The approach however did not mentioned on solution or steps taken when the ambiguity and overlap between different ideologies occurs.

**Reproducibility:**

3: Could reproduce the results with some difficulty. The settings of parameters are underspecified or subjectively determined; the training/evaluation data are not widely available.

**Reviewer Confidence:**

4: Quite sure. I tried to check the important points carefully. It's unlikely, though conceivable, that I missed something that should affect my ratings.

---

> ### Author Rebuttal · Authors · 2023-08-28
>
> We appreciate your thoughtful review. We address the comments as follows.
>
> > **Q1: Ideologies are often complex and can have multiple dimensions. The approach however did not mentioned on solution or steps taken when the ambiguity and overlap between different ideologies occurs.**
>
> **A1:** We agree that ideologies are often complex and can have multiple dimensions. With this in mind, we take thorough considerations during the design of the schema to minimize the ambiguity and overlap between different ideologies. As described in Section 2, we conduct research in social sciences and refer to a series of scales that categorize political attitudes based on one or more facets (e.g., the dual-axis model of ideology). We also consider the characteristics of contemporary social issues (e.g., the gender ideology in women’s rights) to describe the existing variation in ideology. Based on this, we propose the schema. The five domains in the schema reflect **different aspects** of society and the 12 facets are **clearly defined** and have **clear boundaries**, as shown in Appendix (the definitions of 12 facets will be added in next paper revision). Therefore, by careful design of the schema, we minimize the ambiguity and overlap between different ideologies.
>
> In addition, during data annotation, we try to minimize annotators' comprehension bias by carrying out a strict workflow. Before formal annotation, we provide thorough training for annotators and conduct multiple rounds of annotation trials. After each trial, we collect questions from annotators and discuss frequent inconsistencies, based on which, we retrain annotators and revise the schema if necessary. Each tweet is annotated by 3 random annotators and a strict agreement check is performed. The final average inter-annotator agreement score (Krippendorff’s alpha), which is 83.1 (Table 4), also demonstrates that the ambiguity and overlap between different ideologies are reduced effectively. The above are discussed in detail in Section 3.2 (Data Annotation) and 3.4 (Quality Control).
>
> >**Q2: How do you avoid/ reduce the bias when measuring the ideology?**
>
> **A2:** Please refer to Answer 1.
>
> >**Q3: Are all the 56 annotators in the article from graduate school?**
>
> **A3:** Yes. The 56 annotators are all from graduate school, including schools of public management, social science and communication. And we paid them handsome wages.
>
> >**Q4: (Reproducibility) The settings of parameters are underspecified or subjectively determined; the training/evaluation data are not widely available.**
>
> **A4:** Thank you for pointing out this issue. We tune the hyperparameters based on the development set. We will add the parameter setting in Section 4 in next paper revision. The ways of data split are described in Section 4.1 and we will provide our division of data in the supplement.

---

### Official Review · Reviewer_6B2A · 2023-08-12

**Typos Grammar Style And Presentation Improvements:** 1. While we know there are 12 facets,…
**Soundness:** 4

**Excitement:**

4: Strong: This paper deepens the understanding of some phenomenon or lowers the barriers to an existing research direction.

**Paper Topic And Main Contributions:**

The paper constructs a multifaceted ideology detection (MID) task and dataset based on 12K English Twitter posts. Ideology is split into five main domains: Culture, Diplomacy, Economy, Politics, Society. Within each domain, it is split further into petals, which is then split into left, center & right. Their MITweet composes of 14 set of different topics, with each having at least 300 tweets annotated.

**Questions For The Authors:**

Q1. How is the spread score and heat score calculated (for cleaning the tweets)?

**Reasons To Accept:**

1. A well-annotated and designed dataset with an average of 83.1 for Krippendorff's alpha.
2. A valid experiment setup and reporting of both the average and standard deviation of the F1 score.
3. Show that their approach is generalizable as their model with zero & few-shot setting on new topics still perform well.

**Reasons To Reject:**

Weakness: While the dataset is manually annotated, there is no external measure to determine whether the ideology detected matches. A case study on a specific topic could help.

**Reproducibility:**

4: Could mostly reproduce the results, but there may be some variation because of sample variance or minor variations in their interpretation of the protocol or method.

**Reviewer Confidence:**

3: Pretty sure, but there's a chance I missed something. Although I have a good feel for this area in general, I did not carefully check the paper's details, e.g., the math, experimental design, or novelty.

---

> ### Author Rebuttal · Authors · 2023-08-28
>
> We appreciate your thoughtful review and recognition of our work. We address the comments as follows.
>
> > **Q1: How is the spread score and heat score calculated (for cleaning the tweets)?**
>
> **A1:** Due to considerations of paper length and structure, we only provide a brief description in Sec. 3.1. Here are the details.
>
> For each tweet, the spread score is calculated by weighting the number of likes, comments and retweets. We also consider the influence of the tweet author and calculate the user score by weighting the number of tweets, followees and followers of the author. Then the final heat score is obtained by weighting the spread score and user score.
>
> \begin{align}
> \text{spread score} &= \mathrm{\\# likes * \alpha_1 + \\#comments * \alpha_2  + \\#retweets * \alpha_3,}  \\\\
> \text{user score} &= \mathrm{\\#tweets *\beta_1 + \\#followees *\beta_2  + \\#followers *\beta_3,}  \\\\
> \text{heat score} &= \mathrm{spread score * \mu_1  + user score * \mu_2.}
> \end{align}
>
> During data collection, we set $\alpha_1$=0.3, $\alpha_2$=0.6, $\alpha_3$=0.1, $\beta_1$=0.3, $\beta_2$=0.1, $\beta_3$=0.6 and $\mu_1$=0.6, $\mu_2$=0.4. We remove tweets with heat scores below a threshold which is adjusted manually according to the topic heat on Twitter.
>
> We will add the above to the appendix in next paper revision.
>
> > **Q2: (Presentation Improvements) While we know there are 12 facets, it is mainly abbreviated (in Table 3) and other mentions. It's hard to pinpoint which facet is which, until the reader reaches the appendix.**
>
> **A2:** Thank you for pointing out this issue. We will modify Table 3 and add the full name of each facet in next paper revision. Although the full names have been shown in Figure 1(a), we acknowledge that the figure is mainly used to visually show the overall framework of the schema, and it is a little difficult to find the details of each facet.
>
> > **Q3: (Weakness) While the dataset is manually annotated, there is no external measure to determine whether the ideology detected matches. A case study on a specific topic could help.**
>
> **A3:** Thank you for your constructive suggestion.
>
> With well-designed schema and strict annotation workflow, the relevance and ideology distribution of MITweet is generally consistent with the sociological characteristics of each topic. Take the topic of BLM as an example.
>
> BLM (Black Lives Matter) is a social and civil rights movement. It advocates for the rights and equality of Black communities in various aspects of society, including law enforcement, criminal justice, education, employment, and healthcare. The movement emerged in 2013 in response to the acquittal of Trayvon Martin's killer and gained significant momentum following the high-profile deaths of other unarmed Black individuals, such as Michael Brown and Eric Garner. Therefore, BLM is mainly related to Social Development, Justice Orientation, Personal Right (Society domain) and Ethical Pursuit (Culture domain) in the schema. In addition, the nature of BLM, which is the pursuit of black rights, racial equality and social reform, suggests that it stands for Revolutionism (Left) in Social Development, Result Justice (Left) in Justice Orientation, Individual Right (Right) in Personal Right and Ethical Liberalism (Left) in Ethical Pursuit.
>
> The topic distribution of MITweet is shown in Table 11 in Appendix, we can see that the label distribution of BLM topic is consistent with the analysis mentioned above. The facets with the largest number of related tweets are Justice Orientation, Social Development, Personal Right and Ethical Pursuit, accounting for 88.4%, 36.5%, 23.9% and 5.4% of all BLM tweets, respectively. The ideology distribution also aligns with the sociological characteristics of BLM. Most tweets related to Ethical Pursuit (81.6%), Social Development (80.6%) and Justice Orientation (89.0%) are left-leaning in their respective facets. And 87.6% of tweets related to Personal Right are right-leaning in this facet.
>
> We will add the above in Section 3.3 in next paper revision.
>
> > **Q4: (Reproducibility) The settings of parameters are underspecified or subjectively determined; the training/evaluation data are not widely available.**
>
> **A4:** Thank you for pointing out this issue. We tune the hyperparameters based on the development set. We will add the parameter setting in Section 4 in next paper revision. The ways of data split are described in Section 4.1 and we will provide our division of data in the supplement.

---

### Meta-Review · Area_Chair_ZMyK · 2023-09-22

**Recommendation:** 5

**Metareview:**

This study introduces a multifaceted ideology detection dataset comprising 12K English tweets. In addition to the dataset, the research presents benchmark results using various pre-trained language models.

All reviewers acknowledged the value of the dataset, though they had a few minor concerns. The authors addressed these concerns in their rebuttal. It would be beneficial to incorporate these responses into the paper to provide greater clarity for readers.

---

### Decision · Program_Chairs · 2023-10-07

**Decision:**

Accept-Main

**Comment:**

This study introduces a multifaceted ideology detection dataset comprising 12K English tweets. In addition to the dataset, the research presents benchmark results using various pre-trained language models.

All reviewers acknowledged the value of the dataset, though they had a few minor concerns. The authors addressed these concerns in their rebuttal. It would be beneficial to incorporate these responses into the paper to provide greater clarity for readers.